# Real-Time Interference Artifacts Suppression in Array of ToF Sensors [note 1]

**DOI:** 10.3390/s20133701

**Published:** 2020-07-02

**Authors:** Jozef Volak, Jakub Bajzik, Silvia Janisova, Dusan Koniar, Libor Hargas

**Affiliations:** Department of Mechatronics and Electronics, University of Žilina, 010 26 Žilina, Slovakia; jozef.volak@feit.uniza.sk (J.V.); jakub.bajzik@feit.uniza.sk (J.B.); silvia.janisova@feit.uniza.sk (S.J.); libor.hargas@feit.uniza.sk (L.H.)

**Keywords:** 3D imaging, depth sensors, multi-camera interference, interference suppression

## Abstract

Time of Flight (ToF) sensors are the source of various errors, including the multi-camera interference artifact caused by the parallel scanning mode of the sensors. This paper presents the novel Importance Map Based Median filtration algorithm for interference artifacts suppression, as the potential 3D filtration method. The approach is based on the processing of multiple depth frames, using the extraction of the interference region and application of the interpolation. Considering the limitations and good functionalities of proposed algorithm, the combination with some standard methods was suggested. Performance of the algorithm was evaluated on the dataset consisting of the real-world objects with different texture and morphology against popular filtering methods based on neural networks and statistics.

## 1. Introduction

Due to the demand in the last decades, there are several types of depth cameras available on the market. They use the different operation principles of depth sensors, that allow reconstructing the geometric properties of the environment. Often in addition to the depth of the scene, also the radiometric information about the scene is obtained by a coupled RGB sensor [1]. The popular principle for determining the depths of scene-points is called Time of Flight (ToF). With the advantage of non-invasive nature and eye-safety of ToF cameras, medical research and imaging is the potential area of use.

However, besides all advantages, ToF cameras are subject to a large variety of measurement error sources. The number of investigations of these errors were reported and have shown that they are caused by sensor parameters and properties or environment configuration [2]. Essential ToF camera errors include multi-path (MPI) and multi-camera interference (MCI).

In our work, we deal with the analysis of MCI occurs due to the reception of a reflected modulated signal generated from another ToF camera. We also propose the filtering method of extreme values generated by the interference of multiple ToF cameras. This method is designed for object scanning applications where the cameras are directed to one common point. In comparison with other hardware-based interference suppression methods, the proposed algorithm is universal and does not require physical interventions, such as the frequency modulation of the IR light source. The disadvantage of a software-based solution is that there is a possible loss of non-corrupted data. Therefore, we aim to maximize the amount of true interference data removed while retaining as much non-corrupted data as possible. The filtering algorithm is based on the combination of temporal filtering and interpolation of areas damaged by interference. It is possible to recover the 3D model of an object from its multi-view images [3]. Our algorithm is able to reconstruct the damaged data by interpolation from one view. The algorithm should filter the data precisely, which means that the useful points should be preserved. The performance of the algorithm is evaluated on the interference-corrupted point cloud model via comparison with other popular algorithms (Statistical Outlier Removal, Radius Outlier Removal, and PointCleanNet).

## 2. Related Work

Multi-path interference is one of the basic ToF camera errors. The assumption that the light returned to each pixel of the sensor comes only from a single position in the scene is not practically possible and correct. The reason is that the light can travel multiple paths. The well known is the scene based MPI, where the light scattering of multiple points in the scene arrives in the same pixel of the sensor is caused by inhomogeneities of the scene; or intra-camera multi-path interference caused by the light refraction and reflection of an imaging lens and aperture. This interference is described in detail in References [4,5,6,7].

MCI occurs in multi-camera systems where the cameras operate with the light signal of the same modulation frequency and are directed to a common point. This interference is due to the reception of a reflected modulated signal generated from another ToF camera. The depth map error generated by MCI is much more significant than MPI [4].

One of the still remaining problems of ToF cameras are motion artifacts in dynamic scenes, that may lead to false values in the measured object. Creating the most accurate models of dynamic objects such as the pediatric patient is dependent on the scanning time. With a longer time needed for scanning, the motion artifacts and motion errors are introduced into the resulting model [8]. In general, motion artifacts can be minimized with a shorter image acquisition time. Placing the demands on the fast image acquisition excludes the usage of single-camera scanning for such an application with children. A possible solution is to use a multi-camera system that works in parallel mode and collects the data from multiple viewing angles. Although, if we want to use the multi-camera system, it is necessary to reduce the impact of MCI on the generated 3D models.

One concept of the multi-camera system uses a sequential mode where the cameras work at certain synchronized intervals (in one time only one camera is active). It is the opposite method of parallel cooperation of cameras. However, this method is not suitable for the scanning of dynamic objects. Another option is to use cameras that offer multiple modulation frequencies of emitted light. Many recent works dealing with the problem of multi-camera interference state that ToF-cameras must not be operated using the same modulation frequency as a solution [9]. They rely on orthogonal functions, such as sinusoids of different frequency modulation. Although it does not solve the problem if we are limited by the camera’s availability or price. There is also software complication with the connection of different types of cameras in one application [10,11,12].

Using the least-squares estimation of non-interference signal (LSENS), it is possible to restore depth and amplitude information from serious interference. This method is based on the analysis of the statistical properties of signal interference [13]. However, the interfering signal must have a positive and negative fluctuation of the sinusoidal function. In situations where the cameras are directed against each other, the fluctuation of the signal may be extremely high. In this case, the LSENS method will not be effective. The method for MCI error reduction that was proposed by Lee [14] is a Stochastic Exposure Coding (SEC), which includes dividing a camera’s integration time into multiple slots and switching the camera off and on stochastically during each slot. However, the experiment was not carried out with dynamic objects.

## 3. Materials and Methods

### 3.1. Time of Flight Camera Measurement Model

ToF cameras are optical sensors that provide information about the depth of the scene. They contain an active light source that generates an amplitude modulated signal. The signal may have a continuous or impulse character. Most of the ToF cameras emit amplitude modulated continuous wave (AMCW) with a frequency near IR for illumination of the scene [15]. Depth measurement is based on the amplitude measurement of the phase shift of the transmitted and received modulated signal (Figure 1). The depth information for each pixel can be calculated by the synchronous demodulation of the received modulated light in the detector. The demodulation can be performed by interleaving with the original modulated signal. This process is called cross-correlation. In general, the correlation function c(τ) is defined as follows:(1)$$c\left(\tau \right)=s\left(t\right)⊗g\left(t\right)=\lim_{T\to \infty }\frac{1}{T}\int_{-\frac{T}{2}}^{\frac{T}{2}}s(t)\;\cdot \;g(t+\tau )dt,$$
where s(t) is the received optical signal and the g(t) is transmitted (original) signal. Using specific functions for ToF cameras the equations are:(2)g(t)=cosωt
(3)s(t)=1+a·cos(ωt−φ)
(4)$$\begin{array}{c} c\left(\tau \right)=\varphi_{sg}\left(\tau \right)=\frac{a}{2}\;\cdot \;cos\left(\varphi +\omega \tau \right), \end{array}$$
where *a* is modulation amplitude and φ is a phase shift. This function is calculated for four different ωt arguments that are shifted from 0 by 90°. The received signal is mostly superimposed on a background image, which requires adding the offset *b* to the correlation function:(5)$$\begin{array}{cc} \; & C(\tau )=c(\tau )+b \end{array}$$
(6)$$\begin{array}{cc} \; & C\left(\tau_{0}\right)=c\left(\tau_{0}\right)+b=\frac{a}{2}\;\cdot \;cos\left(\varphi \right)+b\\ \; & C\left(\tau_{1}\right)=c\left(\tau_{1}\right)+b=-\frac{a}{2}\;\cdot \;sin\left(\varphi \right)+b\\ \; & C\left(\tau_{2}\right)=c\left(\tau_{2}\right)+b=-\frac{a}{2}\;\cdot \;cos\left(\varphi \right)+b\\ \; & C\left(\tau_{3}\right)=c\left(\tau_{3}\right)+b=\frac{a}{2}\;\cdot \;sin\left(\varphi \right)+b. \end{array}$$

With these four selected points it is possible to calculate the correlation function and determine the phase φ and amplitude *a* of s(t):(7)$$\begin{array}{cc} \; & \varphi =atan\left[\frac{C\left(\tau_{3}\right)-C\left(\tau_{1}\right)}{C\left(\tau_{0}\right)-C\left(\tau_{2}\right)}\right], \end{array}$$
(8)$$\begin{array}{cc} \; & a=\frac{\sqrt{[C\left(\tau_{3}\right)-C\left(\tau_{1}\right){]}^{2}+{\left[C\left(\tau_{0}\right)-C\left(\tau_{2}\right)\right]}^{2}}}{2}. \end{array}$$

The depth *d* is calculated by the following equation:(9)$$\begin{array}{cc} \; & d=\frac{c\;\cdot \;\varphi }{2\;\cdot \;2\pi f}, \end{array}$$
where *c* is the speed of light and *f* is the IR modulation frequency [16].

### 3.2. Multi-Camera Interference Model

In a multi-view ToF camera system each camera uses the same modulation frequency and IR wavelength, so the received signals interact with one another. In the experimental topology of sensors, we use three ToF sensors S1–S3 that are static and placed in the scanning cabin. This system is shown in Figure 2, which describes the interrelationships between the individual cameras and the object.

Each camera generates a signal g(t) and receives the reflected signal s(t) which represents the connection of the generated sub-signals from all cameras *S*. These signals can be described as follows:(10)$$\begin{array}{cc} \; & s_{1}\left(t\right)=s_{11}\left(t\right)+s_{12}\left(t\right)+s_{13}\left(t\right)\\ \; & s_{2}\left(t\right)=s_{21}\left(t\right)+s_{22}\left(t\right)+s_{23}\left(t\right)\\ \; & s_{3}\left(t\right)=s_{31}\left(t\right)+s_{32}\left(t\right)+s_{33}\left(t\right). \end{array}$$

Each sub-signal can be written as:
(11)$$\begin{array}{cc} \; & s_{xy}\left(t\right)=b_{xy}+a_{xy}\;\cdot \;cos(\omega t-\varphi_{xy}). \end{array}$$

The *x* index represents the target camera and index *y* source camera. Next equations serve as an example of the computational model for one signal $s_{1}$. For rest of cameras the mathematical model is derived the same way:(12)$$\begin{array}{cc} \; & s_{1}\left(t\right)=b_{11}+a_{11}\;\cdot \;cos\left(\omega t-\varphi_{11}\right)+b_{12}+a_{12}\;\cdot \;cos\left(\omega t-\varphi_{12}\right)\\ \; & +b_{13}+a_{13}\;\cdot \;cos\left(\omega t-\varphi_{13}\right)={\tilde{b}}_{1}+{\tilde{a}}_{1}\;\cdot \;cos\left(\omega t-\tilde{\varphi_{1}}\right). \end{array}$$

After 4-phase correlation of interfering signals, Equations (Equation 7) and (Equation 8) transform to the following form:(13)$$\begin{array}{cc} \; & {\tilde{\varphi }}_{1}=atan\left[\frac{a_{11}sin\left(\tau_{11}\right)+a_{12}sin\left(\tau_{12}\right)+a_{13}sin\left(\tau_{13}\right)}{a_{11}cos\left(\tau_{11}\right)+a_{12}cos\left(\tau_{12}\right)+a_{13}sin\left(\tau_{13}\right)}\right], \end{array}$$
(14)$$\begin{array}{cc} \; & \tilde{a_{1}}=\sqrt{\frac{a_{11}^{2}+a_{12}^{2}+a_{13}^{2}+2\bar{a}}{2}}, \end{array}$$
where:(15)$$\begin{array}{cc} \; & \bar{a}=a_{11}a_{12}\left[sin\left(\varphi_{11}+\varphi_{12}\right)+cos\left(\varphi_{11}+\varphi_{12}\right)\right]+\\ \; & a_{11}a_{13}\left[sin\left(\varphi_{11}+\varphi_{13}\right)+cos\left(\varphi_{11}+\varphi_{13}\right)\right]+\\ \; & a_{12}a_{13}\left[sin\left(\varphi_{12}+\varphi_{13}\right)+cos\left(\varphi_{12}+\varphi_{13}\right)\right]. \end{array}$$

This mixing of signals from different cameras causes significant measurement errors and therefore the output depth map contains artifacts [13].

## 4. Multi-Camera System Interference Analysis

Equations (Equation 13) and (Equation 14) are valid when each received signal $s_{x}\left(t\right)$ is the combination of the sub-signals of all the cameras S1–S3, where *x* is going from 1 to 3. The interactions of the cameras that are located according to Figure 2 depends on the location of the object in the scene being scanned. If the object is large enough, it can prevent interference with the $s_{32}$ and $s_{23}$ interference sub-signals. Also, if the object is too distant, the S1 interference between the other cameras disappears. The shape of the object also affects interference. When scanning dynamic objects, the effect of interference on output images may change, so image filtering must be responsive to changes (adaptive).

ToF cameras provide RGB, IR and depth images. The depth image encodes the distance of the scanned surface from the sensor to the grayscale image. Thus, the depth image contains 3D information in the 2D plane of the image. The value of individual pixels (32-bit float) represents absolute distance. Having this information, it is possible to reproject the scanned scene, where internal and external parameters of RGB and IR cameras must be known.

Interference affects only the IR image (not RGB) and subsequently, interference artifact occurs in the depth image. In Figure 3 it is possible to see IR and depth image with and without interference. In the depth image, interference is manifested as extreme values considering values from the nearest surrounding. This fact causes some deformities in the 3D reconstructed model. Figure 4 shows the difference between 3D models and points clouds reconstructed from depth images with and without interference.

## 5. Methodology

There are several overview articles dealing with filtration and restoration techniques used in 3D imaging and scanning. From all of them, Reference [17] seems to be very useful, because it contains a review of approximately 40 methods and categorizes them into several classes. Many of these methods are applicable directly to 3D point clouds.

In our case, we considered the interference as a specific place in the signal, where the points are extremely distant from the real surface. Since outliers appear as isolated points with extreme values, interference areas look like specific continuous clusters, as we can see later in Figure 15. The interference suppression means removing these interference points while retaining the useful points. In this paper, we propose several approaches described in the following section.

### 5.1. Experimental Setup

To verify the functionality of the selected filtration methods, we performed an experiment under real conditions on a static object. Three ToF cameras of type Microsoft Kinect v2 were used in the experiment. These cameras were placed in the scanning cabin, the design of which is shown in Figure 5d. In the center of the cabin, the model object is located in the range from 60 cm to 80 cm from the sensors. The object was represented by a plastic head with a relatively complex shape. The size of the object corresponds to normal dimensions of a child aged 5–10 years. The RGB images received from each camera are shown in Figure 5a, the depth frames of the scanned scene without interference (sequentially working cameras) for the individual cameras are shown in Figure 5b. These depth maps were used as reference data with no interference artifacts. Figure 5c is a reconstructed 3D model of a static object.

To objectively evaluate the performance of interference suppression we need to know about every point in the dataset, whether the point is true interference or not. The data distorted by interference were obtained in the parallel scanning mode of cameras. For obtaining ground truth information we used the spatial filtering of interference point clouds against the reference. The reference mesh, shown in Figure 5c, was obtained from the reference point cloud. The overall process of method evaluation is shown in Figure 6.

The steps in the process are described as follows.

Buffering—capturing the several point clouds from a particular view and merging to one cloud.Filtering—applying one of the filtering methods described in Section 5.2.Comparing with reference - computation of Hausdorff distances against reference mesh for a particular view.Spatial filtering—statistical decision, whether the point is interference or not based on its Hausdorff distance from reference. As interference, we consider the statistical outliers from the normal distribution.Analysis—the evaluation of results using the known metrics from ROC analysis.

In this experimental part, we use the following metrics to evaluate the performance of particular methods.

Accuracy reflects the overall algorithm performance, so it also takes into account the true negative predictions (the point is not interference). The accuracy is high when the number of true positive and true negative predictions is large.Recall reflects relatively, how many true interfering points we have removed, without the knowledge about false positives. The recall is high, when the number of true predictions (true interference points filtered) is large, no matter how many useful points we lose.Precision reflects relatively the number of true predictions with respect to the number of false-positive predictions.F1 score is a balance value between recall and precision. This means that the F1 score is not distorted by a large number of true negative predictions and may be considered as decisive.Hausdorff distance measures how far two point sets are from each other. We use the mean value of distances from a point in the cloud to the closest surface in the reference mesh.

### 5.2. Filtering Methods

Actually there are several approaches in the filtering of 3D data. Many of them are based on statistical properties between the given point and its surroundings and there are some based on artificial intelligence. In the following part, we describe our method called Importance Map Based Median (IMBM) filter and also several other standard methods for further result comparison (Statistical Outlier Removal (SOR), Radius Outlier Removal (ROR), PointCleanNet).

#### 5.2.1. SOR Filter

Statistical Outlier Removal is the outlier filtering method based on the statistical analysis on each point’s neighborhood. An outlier can be classified as a misplaced or isolated point or set of points in the point cloud. In the SOR method, we establish the number of points, that will be considered as neighbors (user-defined number of *k* nearest neighbors [18]) and then the mean distance of each point to its neighbors is calculated. The points whose mean distances do not meet the criteria and are outside an interval which is defined by the global distances mean and standard deviation are considered as outliers and trimmed from the dataset [19]. The SOR is a widely used and efficient method implemented as a part of the PCL library, although there is a time processing limitation in large 3D datasets [20].

#### 5.2.2. ROR Filter

Radius Outlier Removal is a quite simple statistical filter. If values of a spherical radius rad and a minimum number of neighbors *k* in the point cloud are given, the filter removes the points which have less than *k* points in a sphere of radius rad centered in the point [21].

#### 5.2.3. POINTCLEANNET

PointCleanNet is a two-stage cleaning deep-learning algorithm that removes outliers and reduces noise in unordered point clouds. The architecture is based on PCPNet (a deep-learning-based approach for estimating local 3D properties in point clouds [22]) with few modifications. In the first step, the outliers are removed. The network uses a novel loss function that removes the outliers without knowledge of the noise characteristics or the information about the surface. In the second step, the correction vectors for the remaining points are estimated. As a deep learning-based method, it takes advantage of high accuracy but the training can be time-consuming and resources-intensive [23]. In the experimental part, we use only the pre-trained model for the removal of the outliers.

#### 5.2.4. Importance Map Based Median Filter

For the suppression and filtering of interference artifacts in the multi-camera system, we proposed an algorithm (see Figure 7). The algorithm is based on interference region extraction and interpolation in a series of images obtained from the camera. The algorithm was proposed mainly for the application of scanning parts of the human body but is not strictly intended only for this application field. The important prerequisite for filtering and interpolation is to have a sufficient number of subsequent depth frames in the buffer. The basic condition is that objects must have the same or very close position in each frame in buffer. This condition is met by fact, that object is static. In the case of the object which can make a move, we have to minimize the buffer size to avoid possible motion artifacts in the resulting depth map.

Firstly, we must set the adequate number of frames for buffer. This number is discussed in experimental results. Each frame going into the buffer is thresholded using threshold values determined by the topology and properties of our system.

If *d(x;y)* is a single pixel of the depth map, the threshold is performed using the following equation:(16)$$\begin{array}{c} d^{\prime }\left(x;y\right)=\left\{\begin{array}{cc} d(x;y), & \mathrm{if}\;d\left(x;y\right)\in <T_{LOW};T_{HIGH}>\\ 0, & \mathrm{otherwise}, \end{array}\right. \end{array}$$
where *d(x;y)* is 32-bit float number and $T_{LOW}$, $T_{HIGH}$ are lower and upper depth threshold values. In our case they are set to 80 and 1300 (these numbers represent real distance in mm). The scanning cabin is shown in Figure 5d and the dimension of the bottom is 1520×1520 mm.

After depth thresholding, the background is removed and images include only the object of interest (Figure 8). This simple step is important and effective because it often removes areas with interference in which pixels tend to take extreme values (overflowing 32-bit stack variable).

In the next step, the thresholded image with a minimum number of non-zero values is selected as a reference image in the buffer. It is because this image with the highest probability contains a minimal number of interfering pixels. The binary map is created from this reference image and each depth image in the buffer is masked with it. In this step, input images are divided into three-pixel regions—background (also including extreme interference regions), interference region pixels (also including some object pixels) and object pixels (also including interference pixels). Interference regions are dilated by structuring element 1×1 square because extreme interference regions are mostly surrounded with lower intensity of interference regions. In this way, the sum of interference pixels is decreased but the number of object pixels in the interference map encreases.

These morphologically processed frames in the buffer are used to compute the median frame and importance map. The median frame pixels are computed as medians of corresponding pixels through all buffer frames, where all zero values are excluded from the median computation. If all the values in number series are zeros, the resulting value is zero. This step is analogous to temporal averaging filtering of image data and also reduces noise processes [24].

Each median value (pixel in median frame) is annotated with information about the number of non-zero values used for median computation. This additive information (2D array) is called importance map (Figure 9a).

Using importance map we have knowledge about the significance of each pixel in the median frame. The small importance value of a given pixel means a high probability of interference occurring at a particular point. For this reason, it seems to be useful to remove such areas by threshold settings.

Pixels in median frame (Figure 9b) with selected importance higher than k·N are preserved, other pixels are set to zero. The *k* represents the importance map threshold and *N* is the size of the buffer. The coefficient *k* can take a range of values from 0 to 1.

The next phase in IMBM filter is an interpolation. Data removed from the median frame could be reconstructed by using bilinear interpolation as an example. The effect and influence of different interpolation are discussed in Reference [25].

Bilinear interpolation can be replaced by another interpolation method (e.g., bicubic, biquadratic...). In the aspect of computation, bilinear transformation is a good compromise between the nearest neighbor method and other more complex interpolation methods. The limitation of interpolation occurs if there is a depth map with a relatively big area of holes. The limitation of interpolation is also the holes situated in the place of a mouth, eye corners, and so forth.

The used interpolation method is adapted for scanning the application of the human head and face. We must avoid and restrict the false reconstruction in the mentioned areas (mouth, eyes...). Therefore we do not perform the interpolation over the data where the depth significantly varies, over the areas between small isolated particles or large areas affected by interference. Interpolation can be disabled in IMBM algorithm or can be replaced by other interpolation methods.

Using the median frame we can subsequently repair input depth maps with interference. In the repairing process, the pixels of depth maps are compared with pixels of the median frame. If the absolute difference value is greater than the selected difference threshold (this value is discussed in the experimental part in the Section 6.4), the pixel in original depth map is replaced by median value (or zero value, depending on the mode of IMBM method). The value of the difference threshold depends on camera noise intensity for the defined distance. If this value is too low, the big number of non-interfering pixels will be replaced by the median value. On the other side, the filter could preserve a large amount of interference in the original image. The filter output can be in the form of the median frame or in the form of non-interfering input images.

Generally, we can say, that applying the proposed algorithm improves depth image (Figure 9c) by reducing flying pixels, noise processes and filling the holes in the resulting 3D model.

## 6. Experimental Results

For a suitable comparison, we need to set up every filtering method for the best performance. Therefore, in the first step, we tested multiple combinations of filter parameters and get the best options for particular filters. As a standard and decisive metric, we consider the F1 score. In this case, the high accuracy may be confusing because of a large number of true negative predictions (number of not filtered points), that distort the results. Also, the low Hausdorff distance in many cases does not mean high filter performance. This means, that using Hausdorff distance, we are not able to see the point, where the filter removes points that do not suffer from interference. The example of the difference in evaluation using mean Hausdorff distance and F1 Score is shown in Figure 10. Measurements of processing-time consumption were performed on Ubuntu OS with processor Intel(R) Core(TM) i5-6440HQ CPU @ 2.60GHz and 16 GB (Zilina, Slovakia) operating memory.

### 6.1. SOR Filtration

The SOR filter allows us to adjust the sdm-parameter (the standard deviation multiplier threshold) and *k*-parameter (the number of points to use for mean distance estimation). Following color maps (Figure 11) shows, how the performance of filtration evaluated as the F1 score is changed depending on parameter selection and buffer size.

Based on the analysis in Figure 11 we obtained the best setup for the SOR filter. The performance seems to have an increasing trend up to the buffer size 5. The ideal combination of parameters for buffer size 5 is sdm 1.5 and *k* 100. At this setup, the algorithm achieves the best F1 score of 0.68, as seen in Table 1.

### 6.2. ROR Filtration

The ROR filter allows us to adjust the radius (the sphere radius used for determining the *k*-nearest neighbors) and minimal pts-parameter (the minimum number of neighbors that a point needs to have in the given search radius in order to be considered an inlier). Following color maps shows, how the performance of filtration evaluated as the F1 score is changed depending on parameter selection and buffer size.

Similar to the previous case, the analysis in Figure 12 shows the best setup for the ROR filter. The performance of filtration is highest for buffer size 5. The ideal combination of parameters is rad 0.004 and pts 10. At this setup, the filter achieves the best F1 score of 0.63, as seen in Table 2.

### 6.3. PointCleanNet Filtration

By using the pre-trained PointCleanNet model for outliers removal, there are no parameters to set up. Therefore, the filtration performance, in this case, is evaluated as depends only on buffer size. The computation time of the filtration is in the order of minutes, so it is irrelevant for the real-time application.

The performance has an increasing trend with buffer size. As seen in Table 3, the best result obtained based on the F1 score is only 0.16 for buffer size 6.

### 6.4. IMBM Filtration

The IMBM filter allows us to adjust the depth threshold (removing the background and extreme interference pixels), importance map threshold (removing regions with low significance) and difference threshold (setting the depth camera noise threshold). Optimal setting of difference threshold for buffer size 2 is shown in Figure 13a and for buffer size 6 in Figure 13b. Based on the analysis of these figures, the best difference threshold for buffer size 2 is value 13 and for buffer size 6 is value 8. This parameter was statically set to 10 in the next section of the experiment.

The proposed filtering method achieves the highest F1 score for buffer size 5. In this point, the improvement in mean Hausdorff distance is 12% (mean distance for the input noised point cloud is 0.001255 m).

### 6.5. Final Comparison

After getting the best setups for particular filtering methods we show the comprehensive comparison of their performance. Figure 14 shows the visualization of Hausdorff distances of the outputs from particular filters for buffer size 5. The buffer size 5 for this comparison was selected based on the best results from Table 4 (except PointCleanNet, which achieves the best results using buffer size 6).

The interfered areas contained most complex details (eyes, chin, nose,...) and in output from SOR and ROR filters, they are lost (holes in models). The output from our IMBM filter recovers corrupted points in the interference area, as seen in Figure 14.

The dataset of several real-world objects was used for the final evaluation of filtering methods. The dataset was collected by authors and contains 23 scans of objects (the ball, the plastic bottle...) captured from various views. The following Table 5 shows the medians of each evaluated metric in the whole dataset. All the results are for buffer size 5.

In many cases, none of the filtering methods is capable of sufficient removal of the outliers as good as interference artifacts. As seen in Figure 15, there remains an amount of interfering points after SOR filtering. On the other hand, the IMBM filtered cloud contains outliers, which reduce the overall F1 score. Therefore, the best results according to the F1 score were achieved by using both filtering methods applied subsequently, as seen in Table 5.

## 7. Discussion

The performance of the proposed algorithm was evaluated in comparison to known methods for point cloud filtering based on how precise they remove the interference artifacts. However, the SOR, ROR and the PointCleanNet filters are developed mainly for denoising and outliers removal [20,21,23,26]. In many cases, the interference artifact has specific statistical attributes and we cannot consider them as outliers. In this paper, we evaluated their performance on the task, for which they are not primarily designed. Nevertheless, based on Hausdorff distance and F1 score, the SOR and ROR filters achieve comparable results with our proposed method for the filtration of interference artifacts. As figured above, our filtering method does not remove the outlying points sufficiently. Therefore, we propose the usage of our algorithm together with some of the outliers filtering methods. Based on results in this research we recommend using filtering methods for specific applications by following their functionalities described in Table 6.

The problematic parts for SOR and ROR filters are corners and edges, that are rounded on the output. As described in Reference [18], the SOR filter perceives points at borders as outlying points. These filters also remove the small areas (clusters of points that are not interference points) [27,28]. As described in Reference [20], SOR filtering is not entirely appropriate for real-time performance.

The IMBM filter can be divided into the filtration part and the interpolation part (which repairs corrupted data). When focusing only on the filtration phase (common for all discussed methods) our method takes the shortest time. With approx. 200 ms for 6-frame buffer computation, we are theoretically able to process 5 full buffers while framing ratio of the sensor can be set to 20–30 fps. In the case of static objects or quasi-static objects we assume this framing frequency as sufficient. Considering the limitations and also good functionalities of individual methods, the suitable solution is the combination of the IMBM method and SOR or ROR. When known filtration methods (SOR, ROR, PCN) removes outliers, it can partially also suppress the interference. On the other hand, our IMBM method removes interference but also it can partially remove outliers. These combinations caused increase in the time consumption and rounding of the corners. However, if the removal of interference and also denoising is requested we accept the cost of limitations mentioned above. The IMBM algorithm we propose does not only aim to remove the interference but also provides the interpolation of missing points. According to this fact, the processing time is increased. The probability that the filter will detect and remove the interference rises with the higher variance in the positions of the interference in between the buffer frames.

## 8. Conclusions

As mentioned in the introduction, the main objective of our research is the analysis and the suppression of multi-camera interference artifacts. The scanning system demands several ToF cameras in its spatial topology, which results in the multi-camera interference error.

In this paper, we describe the theoretical analysis of interference artifacts and propose the novel algorithm for multi-camera interference suppression, as the innovation and extending to standard methods. There are several known outlier removal and denoising methods which are not primarily designed for interference suppression. We conclude that interference and outliers differ in character and we cannot consider them as the same type of data corruption. Therefore, we proposed IMBM algorithm, which achieves the best performance in combination with some of the outliers filtering methods. The proposed combination leads to suppression of noisy artifacts as well as interference, what extends the functionality of known methods. Unlike the other filtering methods, IMBM also provides the interpolation of removed points. Nevertheless, it is still suitable for real-time applications. The specific medical applications focusing on the 3D scanning of head and face are potential areas of usage for the proposed algorithm. In the future work, the research may potentially focus on optimization of the script for the multi-thread processing.

## Figures and Tables

**Figure 1 sensors-20-03701-f001:**
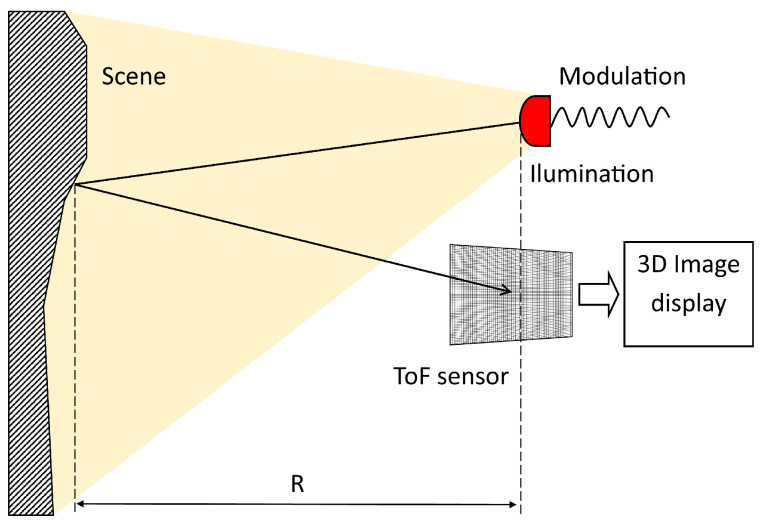
Time of Flight (ToF) camera phase-measurement principle [13].

**Figure 2 sensors-20-03701-f002:**
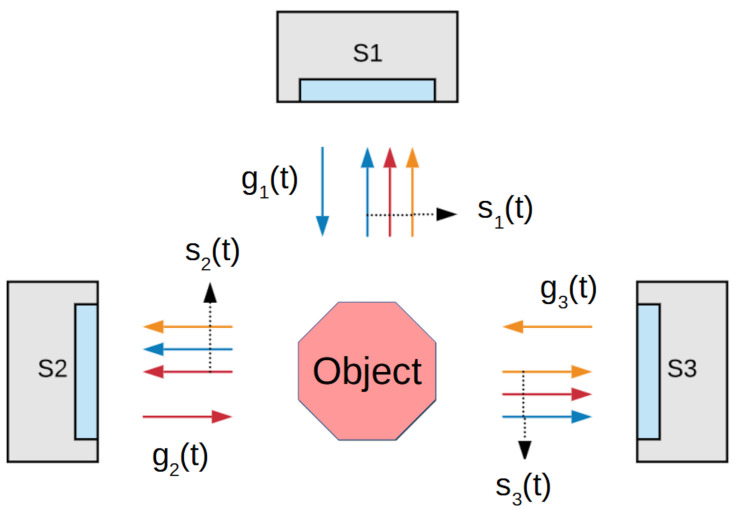
ToF camera phase-measurement principle [13].

**Figure 3 sensors-20-03701-f003:**
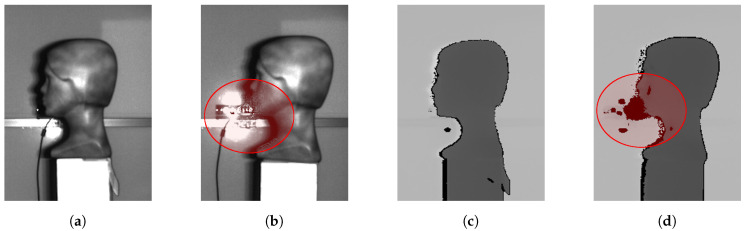
Images from Microsoft Kinect v2 sensor: (**a**) Infrared (IR) frame without interference. (**b**) IR frame with interference. (**c**) Depth image of (**a**) IR image. (**d**) Depth image of (**b**) IR image. Interference is marked with red oval.

**Figure 4 sensors-20-03701-f004:**
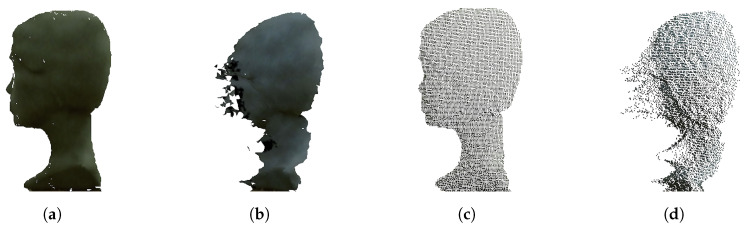
Reconstructed 3D model of static object: (**a**) 3D mesh reconstructed from depth image without interference. (**b**) 3D mesh reconstructed from depth image with interference. (**c**) Point cloud data from depth image without interference. (**d**) Point cloud data from depth image with interference.

**Figure 5 sensors-20-03701-f005:**
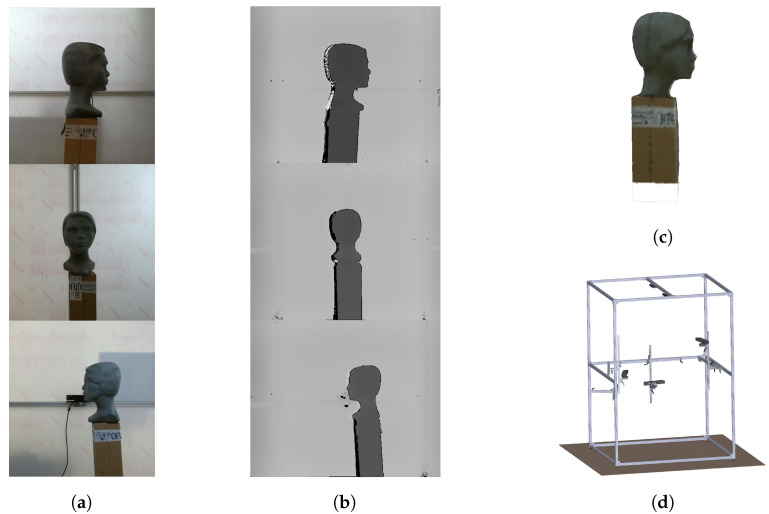
The images from three Kinect v2 ToF camera: (**a**) RGB images. (**b**) Ground truth depth images. (**c**) 3D reconstructed model from ground truth depth images. (**d**) Construction of scanning cabin with placement of cameras, configuration for three cameras.

**Figure 6 sensors-20-03701-f006:**
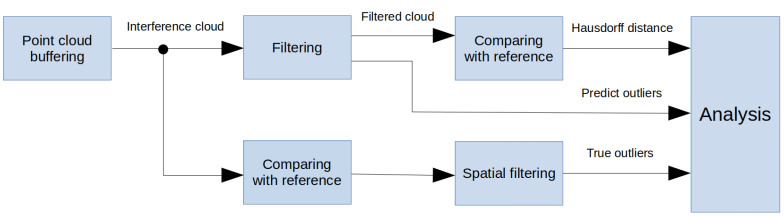
The block diagram of evaluation process.

**Figure 7 sensors-20-03701-f007:**
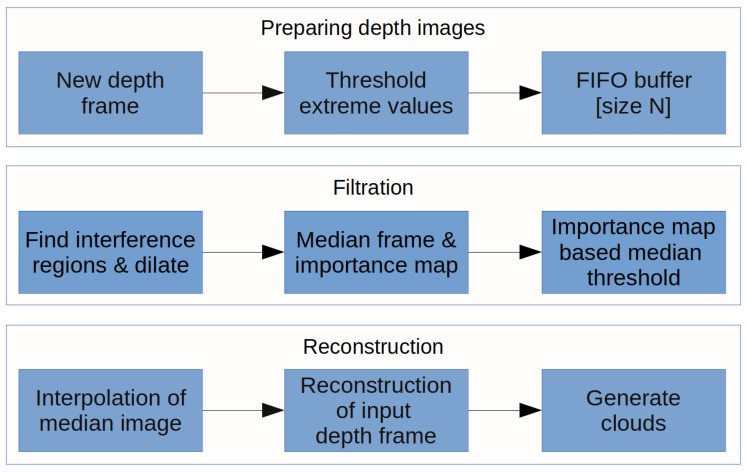
The block diagram of proposed algorithm.

**Figure 8 sensors-20-03701-f008:**
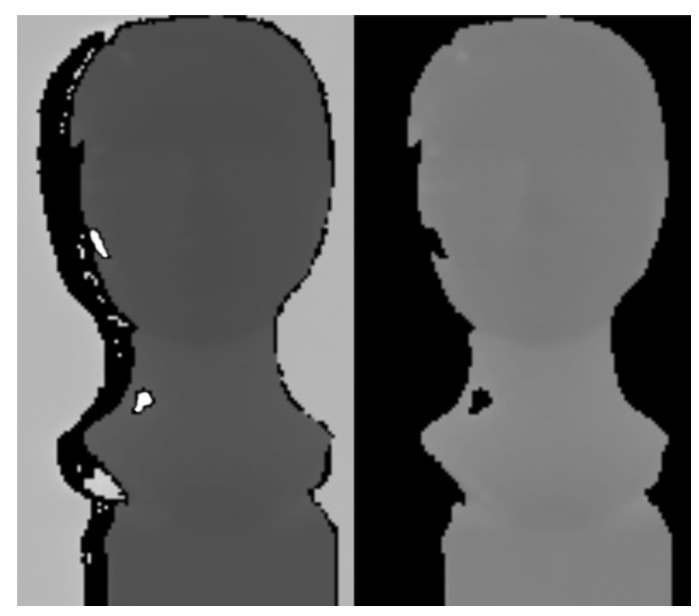
Input depth image with interference. The left image with extreme interference regions before depth thresholding. The right image without extreme interference regions after depth thresholding.

**Figure 9 sensors-20-03701-f009:**
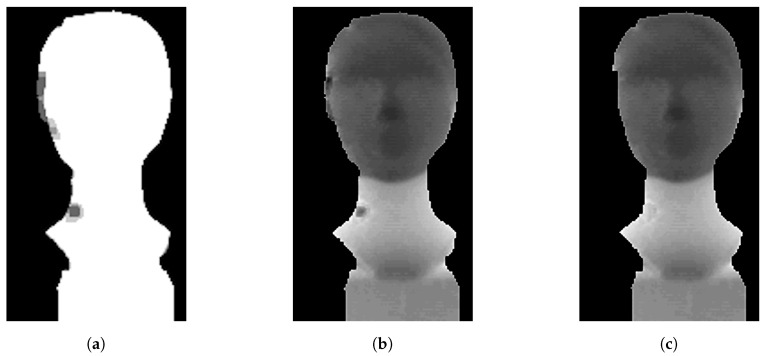
(**a**) Importance map. (**b**) Median. (**c**) Interpolated median depth frame.

**Figure 10 sensors-20-03701-f010:**
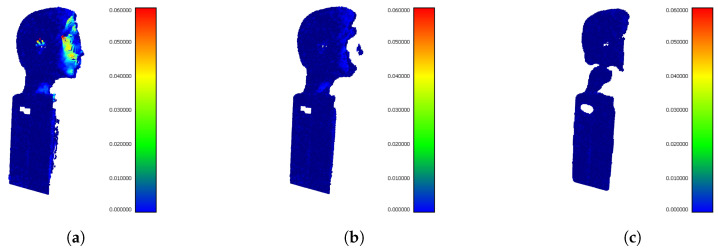
The difference in evaluation using Hausdorff distance and F1 Score. (**a**) Point cloud with interference points. The mean Hausdorf distance is large. (**b**) Precisely filtered point cloud. The mean Hausdorff distance is low and F1 score is large. (**c**) In this case, large number of useful points were lost. The mean Hausdorff distance is still low but also F1 score is low.

**Figure 11 sensors-20-03701-f011:**
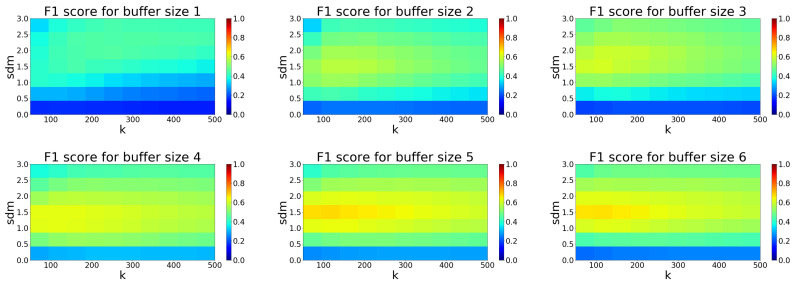
Color map visualization of F1 score dependence on Statistical Outlier Removal (SOR) filter parameters for each size of buffer 1–6.

**Figure 12 sensors-20-03701-f012:**
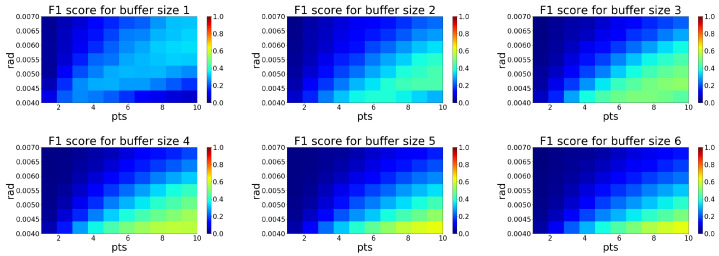
Color map visualization of F1 score dependence on Radius Outlier Removal (ROR) filter parameters for each size of buffer 1–6.

**Figure 13 sensors-20-03701-f013:**
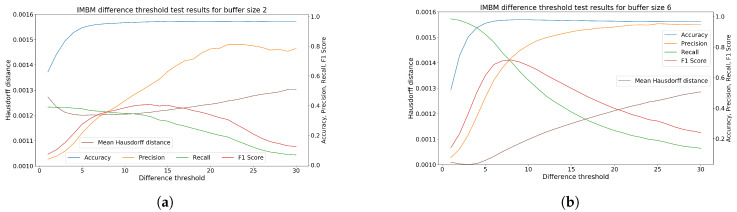
The selection of ideal difference threshold. (**a**) Analysis for buffer size 2. (**b**) Analysis for buffer size 6.

**Figure 14 sensors-20-03701-f014:**
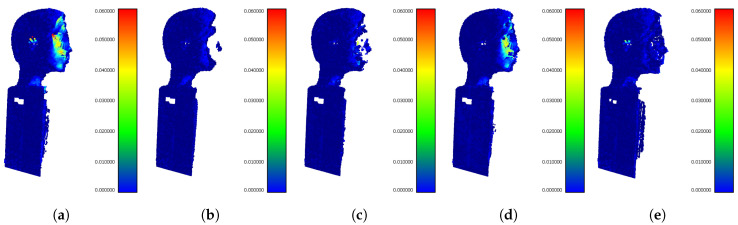
The comprehensive comparison of filtration methods using visualization of distances between filtered clouds and reference mesh. All the results are for buffer size 5. (**a**) The input point cloud without interference suppression. (**b**) The best result using SOR filter with parameters sdm 1.5 and *k* 100. (**c**) The best result using ROR filter with parameters rad 0.004 and pts 10. (**d**) The best result using PointCleanNet for buffer size 6. (**e**) The best result using our IMBM filter.

**Figure 15 sensors-20-03701-f015:**
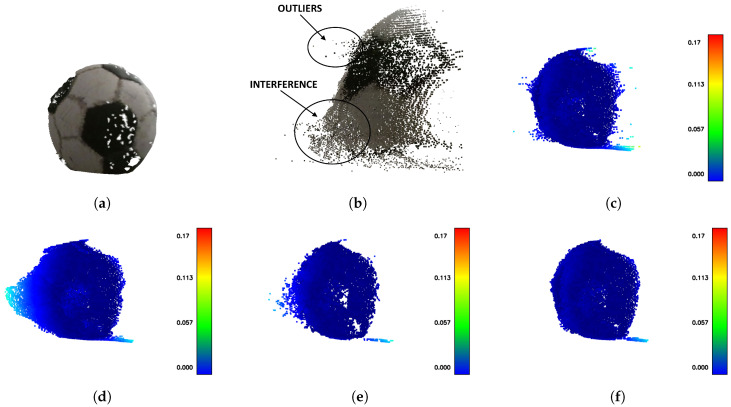
The visualizations of distances for IMBM, SOR and ROR filters. All the results are for buffer size 5. (**a**) The reference (the ball). (**b**) The difference between outliers and interference in the corrupted point cloud. (**c**) The IMBM filter sufficently suppresses the interference but outliers remain. (**d**) The SOR filtered point cloud does not contain outliers but intereference still remains. (**e**) The ROR filtered point cloud does not contain outliers but intereference still remains. (**f**) The best result using SOR and IMBM filter applied subsequently.

**Table 1 sensors-20-03701-t001:** SOR filter best results for multiple buffer sizes.

Buffer Size	Accuracy	Precision	Recall	F1 Score	Mean Hausdorff Distance [m]	Time [ms]
1	0.964001	0.390533	0.538043	0.452571	0.001101	435
2	0.959534	0.474334	0.719212	0.571652	0.001073	454
3	0.964936	0.472317	0.853734	0.608171	0.001025	448
4	0.951624	0.501123	0.839413	0.627583	0.001041	644
5	0.968504	0.615551	0.754316	**0.677905**	0.001062	814
6	0.970065	0.598705	0.76028	0.669887	0.001053	979

**Table 2 sensors-20-03701-t002:** ROR filter best results for multiple buffer sizes.

Buffer Size	Accuracy	Precision	Recall	F1 Score	Mean Hausdorff Distance [m]	Time [ms]
1	0.956411	0.296154	0.418478	0.346847	0.001116	47
2	0.944072	0.352522	0.585222	0.44	0.001104	89
3	0.963341	0.446866	0.631255	0.523293	0.001064	166
4	0.949669	0.487503	0.711546	0.578593	0.001092	167
5	0.962245	0.553508	0.728088	**0.628907**	0.001068	235
6	0.968228	0.590323	0.6689	0.62716	0.001074	325

**Table 3 sensors-20-03701-t003:** PointCleanNet best results for multiple buffer sizes.

Buffer Size	Accuracy	Precision	Recall	F1 Score	Mean Hausdorff Distance [m]
1	0.840448	0.084714	0.486413	0.144297	0.00112
2	0.835066	0.07334	0.291626	0.117205	0.001226
3	0.830422	0.06211	0.30639	0.103283	0.001181
4	0.819131	0.096918	0.327567	0.149579	0.001387
5	0.819513	0.094805	0.363546	0.150391	0.001324
6	0.823568	0.099486	0.424307	**0.16118**	0.001263

**Table 4 sensors-20-03701-t004:** Importance Map Based Median (IMBM) filter best results for multiple buffer sizes.

Buffer Size	Accuracy	Precision	Recall	F1 Score	Mean Hausdorff Distance [m]	Overall Time [ms]	Filtering Time [ms]
2	0.958461	0.433661	0.347783	0.386003	0.001205	179	84
3	0.978235	0.724401	0.511932	0.59991	0.001119	247	116
4	0.950948	0.493889	0.410305	0.448233	0.001229	291	136
5	0.97733	0.815311	0.62583	**0.708114**	0.001114	337	158
6	0.977306	0.807993	0.566555	0.66607	0.001107	426	200

**Table 5 sensors-20-03701-t005:** The medians of each evaluated metric in whole dataset.

Method	Accuracy	Precision	Recall	F1 Score	Mean Hausdorf Distance [m]
SOR	0.936164	0.980778	0.393617	0.555951	0.002393
ROR	0.927161	0.639983	0.734787	0.647839	0.001697
PCN	0.862285	0.371242	0.472615	0.409729	0.00221
IMBM	0.908833	0.673608	0.442942	0.477348	0.002563
IMBM + ROR	0.918448	0.59806	0.820666	0.653113	0.001656
IMBM + SOR	0.943387	0.704448	0.836253	**0.741811**	0.001548

**Table 6 sensors-20-03701-t006:** Functionality of compared filters.

Method	Reducing Interference	Real-Time	Corners/Edger Rounding	Recovering Input Data
SOR	Medium	Medium	Medium	No
ROR	Medium	High	Medium	No
PCN	Low	Low	Medium	Yes (Noise removal)
IMBM	High	High	Low	Yes

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
