# Peer review of "Real-Time Interference Artifacts Suppression in Array of ToF Sensors [Author-notes fn1-sensors-20-03701]"

_sensors, 2020, doi:10.3390/s20133701_

Round 1
Reviewer 1 Report
This study was aimed to reduce interference artifacts in 3D measurements using ToF sensors. The paper fits this journal. However, I think the following points need correction.
- You use a pre-learning model for PCN, but you should train using your target data.
- Real-time processing is featured in your research, but detailed data (calculation time) is not shown.
- c in equation (1) should also be introduced in the manuscript.
- The contents of reference 17 are closely related to your research. A detailed introduction to the literature is needed.
Author Response
Respected reviewer,
thank you for the comments. We bring the step-by-step responses:
Ad 1
In our opinion, training the PCN using our specific data should increase the performance of outliers removal, but not interference removal. On the other hand, to conclude this, we need a huge dataset of point clouds (our data). If you can see in following figure (Figure 1 - see the attachment), the character of interference differs from standard outliers. Since outliers appears as isolated points with extreme values, interference areas look like specific continuous point clusters.
In this case, potential solution can be in training relevant neural network for filtering specific shapes. This could be a good motivation for further research. Actually, there are some neural networks dealing with shapes: PointNet,
RS-CNN…
In the Figure 15 (Final comparison section) we show something like “reciprocity effect“:
- When known filtration method (SOR, ROR…) removes outliers, it can partially also suppress the interference;
- When our IMBM method removes interference, it can partially remove also outliers;
- From this point of view, we see the potential in the combination of filters declared in Discussion.
Figure 15 was redrawn.
Ad 2
Your comment about algorithm timing properties is absolutely correct. We bring a detailed view on timing of our method and these facts are put into the manuscript to Results, Discussion section and in the Table 4 (text color – red).
As real-time behavior of algorithm we consider such algorithm, where one portion of data is processed before the another portion will come. Following another definition [1]: Real-time programs must guarantee response within specified time constraints… In our case, these constraints can be represented by framing ratio of sensors/cameras.
On the other side, we do not strictly declare, that our method is always real-time, but there is a strong potential to become real-time in specific application.
In case of IMBM (our) method:
IMBM can be divided into filtration part and interpolation part (which repairs corrupted data). User of algorithm can choose just filtration of data or both filtration and interpolation (reconstruction).
The timing of single phases (for 6-frame buffer) is as follows:
Filtration part (thresholding of depth maps, morphology, median frame and importance map computation) – approx. 50 – 60 ms.
Reconstruction of depth maps based on median frame and point cloud generating – approx. 150 ms.
Interpolation of data (e.g. bilinear filter) – approx. 200 – 250 ms.
When concentrating only on filtration phase – common for all discussed method in comparison with IMBM: our method needs the shortest time. With approx. 200 ms for 6-frame buffer computation, we are theoretically able to process 5 full buffers and framing ratio of sensor can be set to 20 - 30 fps. In the case of static objects or quasi-static objects we assume this framing frequency as sufficient.
All computations were provided on Linux Ubuntu OS, code was written in C++ (PCL Library), hardware: CPU Intel ®Core i5-6440HQ 2.60 GHZ, 16 GB RAM.
As we discuss in Conclusions (future work), there can be also the place for another code optimization.
[1] Ben-Ari, M., "Principles of Concurrent and Distributed Programming", Prentice Hall, 1990. ISBN 0-13-711821-X. Ch16, Page 164.
Ad 3
Variable c in Equation (1) was introduced to the manuscript.
Ad 4
Reference 17 is a review article dealing with many methods and groups of 3D filtering methods. In our opinion, reference 17 is useful for any reader to bring an overview of mentioned methods and give him a good starting point for work with filtration in 3D. In other hand, our effort is not to create a review article and to repeat the information from this reference 17.
We also reflect on comment of another reviewer that to compare and draw the performance of our method by obtaining the algorithms from another research groups might be hard and sometimes impossible.
Our team focused mainly on references dealing with interference and multi-camera systems.
The English grammar was corrected using Grammarly and consulted with English native speakers.

Reviewer 2 Report
For building a completed 3d model, multiple TOF cameras often are used. But the multi-camera interference will affect the depth from the TOF cameras. The paper proposed a software-based solution to suppress the interference artifacts. The filtering algorithm is proposed based on the combination of the temporal filtering and interpolation of areas damaged by interference.
The experiments show that the proposed IMBM algorithm achieves the best performance in cooperation with other outlier filtering methods.
I think the method is novelty and soundness. And it will be interesting to the readers who are doing the same TOF model reconstruction.
Author Response
Respected reviewer,
thank you for your report which supported us in our research work and publishing results in this article.
Reviewer 3 Report
The paper describes a processing workflow for interference artefacts reduction/removal in multi-range camera applications. The authors provide in the introduction a brief overview of multi-range camera interference connected works and some good review papers. The overall structure of the paper is convincing with a good materials and methods section that can be easily understood also by not expert (in this particular field) readers. The proposed workflow is quite simple but the simplicity of the proposed method is not necessarily a flaw.
In a previous submission of this paper I suggested that the authors should extend their testing dataset, maybe using different models and/or different sensors and/or different sensors setup. After that submission, the paper has improved quite significantly but the authors stated that they are not able, at this time, to provide additional tests on other datasets or compare their performances with other techniques. Since comparisons with other methods are not presented, it is difficult to understand whether the method actually works well. This remains, in my opinion, the only critical flaw of the paper. I understand providing additional datasets or obtain other research groups' algorithms/applications to draw a performance comparison might be hard (and sometimes might be impossible). However, I still think this would considerably improve the reliability of the authors results and claims. Consequently, in this review round, I suggest again the authors to put an additional effort on this specific point.
Author Response
Respected reviewer,
thank you for the comments. We are fully agreeing with your conclusions and we offer following response.
The primary claim of the paper is to analyze the interference artifacts and illustrate the system for interference removal. Our work offers an analysis of interference artifacts and describes the problems associated with software-based suppression. We have tested our algorithm on different models (from our dataset), and evaluated its performance against the traditional filtering methods. We also describe the difference between the interference removal and outlier removal (denoising). We conclude that we cannot consider these forms of data corrupting as the same. Since outliers appear as isolated points with extreme values, interference areas look like specific continuous clusters.
Therefore, as the best solution for 3D data processing we propose the combination of interference removal algorithm and outliers removal algorithm, as seen in Figure 15 and Table 5. We agree with your comment, that we can find many other filtration methods, that can be compared with ours. As we pointed out, based on characteristic properties of interference and outliers we consider that the testing other denoising methods might be not so proper. Any of known standard methods are not primarily focused on interference removal, so we offer the innovation to standard methods.
We are many thanks for such relevant comments for our paper, based on which we can improve the level of our work. The potential way might be the combination of our algorithm with another outlier removal method. The motivation for further research can be also in using some neural networks dealing with specific shape recognition and removal (PointNet, RS-CNN).
We added some new information to paper according to another reviewers (highlighted red) and we hope that these will fit also your requirements.
Round 2
Reviewer 1 Report
All comments have been addressed and this version is acceptable for publication.
Reviewer 3 Report
All comments have been addressed and this version is acceptable for publication.
This manuscript is a resubmission of an earlier submission. The following is a list of the peer review reports and author responses from that submission.
Round 1
Reviewer 1 Report
The paper describes a processing workflow for interference artefacts reduction/removal in multi-range camera applications. Although the authors highlight that the system is designed for a specific application (medical screening for paediatric patients) their results can easily be extended to a wider range of applications where acquisition of dynamic scenes/objects is required. They provide in the introduction a brief overview of multi-range camera interference connected works and some good review papers, but the list of relevant references can surely be extended, being the topic quite common in scientific literature. The overall structure of the paper is convincing with a good materials and methods section that can be easily understood also by not expert (in this particular field) readers. The proposed workflow is quite simple and is based on a simple thresholding algorithm followed by erosion and interpolation of the depth map and a median filtering stage using several consecutive depth maps. The simplicity of the proposed method is not necessarily a flaw, but since comparisons with other methods are not presented, it is difficult to understand whether the method actually works well. In the experimental results section the authors just analyse the performance of the algorithm considering the influence of some algorithm parameters (i.e. the number of depth maps and the accepted ratio of discarded (outlier labelled) data points. In the end, it is really hard, in my opinion, to understand if the methodology actually satisfies the requirements of the proposed application (or any other application) and if represents a significant step-forward in interference reduction in multi-range ToF camera surveys. The English style and language must be strongly revised and improved as in this form the paper, in my opinion, cannot be accepted for publication.
Some minor detailed remarks follows:
Ln 7: In what does a patient-less diagnostics approach consist?
Ln8: In my opinion including references in the abstract is not a good practice: I want as a reader to find out as fast as possible, reading the abstract, if the topic of the paper is of some interest for me. In this case the references can be easily moved in the introduction chapter.
Ln21: The authors should specify that ToF sensor actually do not register colour information but are generally coupled with an RGB sensor that provides the radiometry of the scene.
Ln 27: Suspended sentence: “The advantage of ToF sensors (working in IR band with small power).”
Ln 37: The authors state that MPI (multi-path interferences) occur in specific cases without giving any addition information. It is therefore not clear if such effects are of some importance for their research and application and whether their technique also reduces MPI.
Eq (11): I think also the parameter a in the equation should have xy indices.
Ln 94-100: I did not understand (but that is probably my fault) how the previous equations (in particular eqs. 13 and 14) are used in the proposed workflow.
Figure 4: I think, presenting the data, that using the same order in figure 3 and 4 would be more effective: e.g. img (a) and (c) data without interference and img (b) and (d) with interference (as in figure 3).
Ln 187; I honestly have some concerns about the quality metrics adopted: the authors state that they only considered pixels in the interpolated regions. I suppose the actual quality of the interpolated values is strongly case dependent being connected to the specific shape of the object, the actual overlapping (and thus interfering) region of the range cameras and on the identification and labelling of eroded/removed pixels by the algorithm. What if some interfering pixels are not eroded/removed and thus interpolated? Does the subsequent Hausdorff distance take into account such discrepancies? I’m not saying the selected metrics are wrong: I’m saying the authors should discuss clearly their choices to clarify my (and probably of many other readers) doubts.
Ln 195: the coefficient c1 is repeated while coefficient c2 is never defined (it is surely a typo…)
Author Response
Respected reviewer,
we are very thankful for your comments and criticism which will improve the quality of our paper. Following your remarks, we did the changes and improvements as follow:
Ad Ln 3: The information about the cases how and when the MPI occurs were added to the introduction, as well as the relevant references. We state that we propose the algorithm for filtration of extreme values generated by the multi-camera interference. It is possible that the developed filtration method also reduces the errors caused by MPI, but we do not distinguish and evaluate the success rate of the algorithm on individual errors of the ToF camera (MPI, MCI, ..).
Ad Ln 7: The patient-less approach expression has the following meaning: Because each medical diagnostics and investigation is stressful for the patient (especially pediatric one), we try to minimize the time of interaction between subject and diagnostic tool. In our case, this interaction can be in seconds - just for scanning and after this procedure, the patient is no longer needed for further data processing. Measurement is performed via computer model, in spite of the conventional method (based on paper sleep questionnaire), where a medical specialist interacts with the patient for a longer time and measures the selected features manually. So patient-less approach expression has the sense in the phase of measuring the cranio-facial features and dimensions.
Ad Ln 8 - Ln 21 - Ln 27 - Ln 37 - Eq 11 - Figure 4: corrected
Ad Ln 94-100: The mathematical model is used only for an illustration of the situation. The equations are not used in our filtration method (algorithm).
Ad Ln 187: Considering your comment, for measuring and evaluating the quality of our method and for comparison with standard and used techniques, we added some new measurements and objective metrics to do so. We added statistic-based SOR, ROR methods, deep-learning and AI-based method PointCleanNet and compared and discussed the methods together.
In this stage of the research it is too early to state that the novel filtration method will satisfy the requirements of the proposed application for scanning of the human heads and faces. In the near future we aim to test the algorithm on the large data set and apply it on the dynamic object of interest. However we can certainly present the algorithm as a strong suppression method for multi- camera interference error, which was the goal of our paper.
All the comments of formal character, language and grammatical mistakes were corrected directly in paper.
Authors

Reviewer 2 Report
The paper presented is interesting and it contributes to the artifacts suppression problem in an array of ToF sensors. I have some comments to improve the manuscript.
The state of art is incomplete, please add recent works about this problem.
The Introduction must be addressed to the real-time interference artifacts proposal instead of the application to pediatric patients, the reader can be confused about the main objective of the work.
The manuscript can be improved by a minor spell checking, there are few typos, grammatical errors and writing problems:
- Line 27: "The advantage of ToF sensors (working un IR band with small power). In comparison with many laser scanners, ToF devices are eyesafe." There isn't continuity of the idea.
- Line 32: "(so)obtaining..."
- Line 121: "...(see Figure 5 based on finding...."
- Line 133: "...is 32bit float number..."
There is missing the definition of Eq. (1), as well as the operator used.
Line 80: There is missing the definition of the experiment.
Define "the importance map" theory before using it.
Line 152: Who are the variables k and N?
Line 154: Can you perform a sensitivity analysis to define the optimal values of k?
Line 195: There is missing the definition of c2, and why do you set k1 and k2 in these values?
In Figures 11-14 can you highlight the best regions for the optimal parameters?
Line 224:Can you argue about the real eye-safe in infrared devices? Is there not eye damage to use these devices?
In Fig. 15 add the number of seconds on the x-axis.
The discussion section is hard to read, can you add some tables to improve the reading?
Author Response
Respected reviewer,
we are very thankful for your comments and criticism which will improve the quality of our paper. Following your remarks, we did these changes and improvements:
Ad state of art and introduction part: Some new information and references dealing with our topic were added to the article.
Minor spell checking, some comments of formal character, grammar and language were corrected. Also figures were changed due to related comments.
Ad Line 27 - 32 - 121- 133: corrected
Ad Line 80: experiment in the meaning of the experimental topology of used sensors.
The importance map is the additive information in the format of 2D, where ach median value (pixel in median frame) is annotated with information about the number of non-zero values used for median computation
Ad Line 152: The coefficient k represents the importance map threshold and the coefficient N is the size of the buffer.
Ad Line 154: Analysis of relevant filtration method also other additives were performed and added into the article. We used objective metrics also for comparison of our proposed method with standard and known ones.
As Line 195: Equation is unused in the new version of the article
Figures 11-14: replaced. We present the new settings for the optimal parameters of each filter:
SOR filter- buffer size=5, sdm=1.5 and k=100.
ROR filer - rad=0.004 and pts=10.
PointCleanNet: buffer size=6
IMBM filter - the best difference threshold for buffer size 2 is value 13 and for buffer size 6 is value 8
Fig. 15: replaced
Ad Discussion: due to the completely reworked experimental part and more complex comparison of proposed methods we changed and extended the discussion part. We hope that now this part of the article is much more clear and easy to understand for the reader.
Ad ToF safety: There are some discussions and forums found on the internet. As example:
https://skeptics.stackexchange.com/questions/2625/is-the-kinect-ir-laser-safe.
The amount of power emitted, distance of eye from sensor and time factor are considered as risk factors discussed but the opinions differ from discussion to discussion. In scientific databases the articles dealing with ToFs discuss and compare mostly the applicability and accuracy of sensors. In our opinion (of authors) - the exposition time is very low and distance of head from camera is relatively big. Comparing it with the playing of video games with these sensors for a long time we are sure about safety.
Authors

Round 2
Reviewer 1 Report
The authors have improved quite consistently the paper, in particular working extensively on the english language and style. Overall the quality of the paper is acceptable and most (if not all) my minor remarks have been addressed properly. My main concern about the article, however, is still not appropriately addressed despite the remarkable efforts of the authors. I'll try to be more specific. The work of the authors would be considered complete, consistent and totally convincing either if
a) they want to propose a system designed as a screening tool to obtain an accurate 3D model of a human’s head and face for investigation of the presence of obstructive sleep apnea syndrome in pediatric patients: in this case they should provide some experimental data that shows clearly that their application goals have been reached to some extent.
OR
b) they want to illustrate a general purpose system for interference removal/reduction (that could be furthermore developed for specific needs/applications). This as far as my understanding of the paper is correct is their primary claim. In this case, they should extend their testing dataset, maybe using different models and/or different sensors and/or different sensors setup. I appreciated they extended their comparison adding some new measurements and objective metrics and the subsequent discussion of the results highlight that their method is promising to some extent, but that, in my opinion, is still not enough. In one of their answer the authors say they are aiming testing the algorithm on a larger data set (and also on some dynamic object, but I think that's not so relevant at this stage): I think this would considerably improve the reliability of their results and claims.
Author Response
Dear reviewer,
Thank you for your answer. We are very grateful for such relevant comments for our paper, based on which we can improve the level of our work. We are aware of the weak part of the paper related to the small dataset.
As you write in the comments, the primary claim of the paper is to analyze the interference artifacts and illustrate the system for interference removal. We realize that our system cannot yet be declared as a general-purpose system. Hence, we continue in our efforts to evaluate the performance of the system on a large dataset.
Unfortunately, it is not possible to create such a dataset synthetically and at the moment we are not able to create an extensive dataset suitable for all the methods described. We have already tested our algorithm on different models, where the interference was suppressed. Unfortunately, we have no reference models yet for the evaluation of the performance.
However, we believe that our work offers an analysis of interference artifacts and describes the problems associated with software-based suppression. It also offers a way to evaluate the performance of the system in a relevant way.